# UNIFORM PRIORS FOR DATA-EFFICIENT TRANSFER

## ABSTRACT

Deep Neural Networks have shown great promise on a variety of downstream applications; but their ability to adapt and generalize to new data and tasks remains a challenge. However, the ability to perform few or zero-shot adaptation to novel tasks is important for the scalability and deployment of machine learning models. It is therefore crucial to understand what makes for good, transferable features in deep networks that best allow for such adaptation. In this paper, we shed light on this by showing that features that are most transferable have high uniformity in the embedding space and propose a uniformity regularization scheme that encourages better transfer and feature reuse. We evaluate the regularization on its ability to facilitate adaptation to unseen tasks and data, for which we conduct a thorough experimental study covering four relevant, and distinct domains: few-shot Meta-Learning, Deep Metric Learning, Zero-Shot Domain Adaptation, as well as Out-of-Distribution classification. Across all experiments, we show that uniformity regularization consistently offers benefits over baseline methods and is able to achieve state-of-the-art performance in Deep Metric Learning and Meta-Learning.

## 1 INTRODUCTION

Deep Neural Networks have enabled great success in various machine learning domains such as computer vision (Girshick, 2015; He et al., 2016; Long et al., 2015), natural language processing (Vaswani et al., 2017; Devlin et al., 2018; Brown et al., 2020), decision making (Schulman et al., 2015; 2017; Fujimoto et al., 2018) or in medical applications (Ronneberger et al., 2015; Hesamian et al., 2019). This can be largely attributed to the ability of networks to extract abstract features from data, which, given sufficient data, can effectively generalize to held-out test sets.

However, the degree of generalization scales with the semantic difference between test and training tasks, caused e.g. by domain or distributional shifts between training and test data. Understanding how to achieve generalization under such shifts is an active area of research in fields like Meta-Learning (Snell et al., 2017; Finn et al., 2017; Chen et al., 2020), Deep Metric Learning (DML) (Roth et al., 2020b; Hadsell et al., 2006), Zero-Shot Domain Adaptation (ZSDA) (Tzeng et al., 2017; Kodirov et al., 2015) or low-level vision tasks (Tang et al., 2020). In the few-shot Meta-Learning setting, a meta-learner is tasked to quickly adapt to novel test data given its training experience and a limited labeled data budget; similarly fields like DML and ZSDA study generalization at the limit of such adaptation, where predictions on novel test data are made without any test-time finetuning. Yet, despite the motivational differences, each of these fields require representations to be learned from the training data that allow for better generalization and adaptation to novel tasks and data. Although there exists a large corpus of domain-specific training methods, in this paper we seek to investigate what fundamental properties learned features and feature spaces should have to facilitate such generalization.

Fortunately, recent literature provides pointers towards one such property: the notion of "feature uniformity" for improved generalization. For Unsupervised Representation Learning, Wang & Isola (2020) highlight a link between the uniform distribution of hyperspherical feature representations and the transfer performance in downstream tasks, which has been implicitly adapted in the design of modern contrastive learning methods (Bachman et al., 2019; Tian et al., 2020a;b). Similarly, Roth et al. (2020b) show that for Deep Metric Learning, uniformity in hyperspherical embedding space coverage as well as uniform singular value distribution embedding spaces are strongly connected to zero-shot generalization performance. Both Wang & Isola (2020) and Roth et al. (2020b) link the uniformity in the feature representation space to the preservation of maximal information

and reduced overfitting. This suggests that actively imposing a uniformity prior on learned feature representations should encourage better transfer properties by retaining more information and reducing bias towards training tasks, which in turn facilitate better adaptation to novel tasks. However, while both Wang & Isola (2020) and Roth et al. (2020b) propose methods to incorporate this notion of uniformity, they are defined only for hyperspherical embedding spaces or contrastive learning approaches[1], thus severely limiting the applicability to other domains.

To address these limitations and leverage the benefits of uniformity for any type of novel task and data adaptation for deep neural networks, we propose uniformity regularization, which places a uniform hypercube prior on the learned features space during training, without being limited to the contrastive training approaches or a hyperspherical representation space. Unlike e.g. a multivariate Gaussian, the *uniform* prior puts equal likelihood over the feature space, which then enables the network to make fewer assumptions about the data, limiting model overfitting to the training task. This incentivizes the model to learn more task-agnostic and reusable features, which in turn improve generalization (Raghu et al., 2019). Our *uniformity regularization* follows an adversarial learning framework that allows us to apply our proposed uniformity prior, since a uniform distribution does not have a closed-form divergence minimization scheme. Using this setup, we experimentally demonstrate that *uniformity regularization* aids generalization in zero-shot setups such as Deep Metric Learning, Domain Adaptation, Out-of-Distribution Detection as well as few-shot Meta-Learning. Furthermore, for Deep Metric learning and few-shot Meta-Learning, we are even able to set a new state-of-the-art over benchmark datasets.

Overall, our contributions can be summarized as:

- We propose to perform *uniformity regularization* in the embedding spaces of a deep neural network, using a GAN-like alternating optimization scheme, to increase the transferability of learned features and the ability for better adaptation to novel tasks and data.
- Using our proposed regularization, we achieve strong improvements over baseline methods in Deep Metric Learning, Zero-Shot Domain Adaptation, Out-of-Distribution Detection and Meta-Learning. Furthermore, *uniformity regularization* allows us to set a new state-of-the-art in Meta-Learning on the Meta-Dataset (Triantafillou et al., 2019) as well as in Deep Metric Learning over two benchmark datasets (Welinder et al., 2010; Krause et al., 2013).

## 2 BACKGROUND

### 2.1 GENERATIVE ADVERSARIAL NETWORKS (GANS)

Generative Adversarial Networks (GANs, Goodfellow et al. (2014)) were proposed as a generative model which utilizes an alternative optimization scheme that solves a minimax two-player game between a generator, $G$, and a discriminator, $D$. The generator $G(z)$ is trained to map samples from a prior $z \sim p(z)$ to the target space, while the discriminator is trained to be an arbiter between the target data distribution $p(x)$ and the generator distribution. The generator is trained to *trick* the discriminator into predicting that samples from $G(z)$ actually stem from the target distribution. While many different GAN objectives have been proposed, the standard "Non-Saturating Cost" generator objective as well as the discriminator objective can be written as

$$\mathcal{L}_D = \max_D \mathbb{E}_{z \sim p(z)}[1 - \log D(G(z))] + \mathbb{E}_{x \sim p(x)}[\log D(x)] \qquad (1)$$

$$\mathcal{L}_G = \min_G \mathbb{E}_{z \sim p(z)}[1 - \log D(G(z))] \qquad (2)$$

with $p(z)$ the generator prior and $p(x)$ a defined target distribution (e.g. natural images).

### 2.2 FAST ADAPTATION AND GENERALIZATION

Throughout this work, we use the notion of "fast adaptation" to novel tasks to measure the transferability of learned features, and as such the generalization and adaptation capacities of a model. Fast adaptation has recently been popularized by different meta-learning strategies (Finn et al., 2017; Snell et al., 2017). These methods assume distinct meta-training and meta-testing task distributions,

---

[1]By imposing a Gaussian potential over hyperspherical embedding distances or pairwise sample relations.

where the goal of a meta-learner is to adapt fast to a novel task given limited samples for learning it. Specifically, a few-shot meta-learner is evaluated to perform $n$-way classification given $k$ 'shots', corresponding to $k$ examples taken from $n$ previously unseen classes. Generally, one distinguishes two types of meta-learners: ones requiring $m$ training iterations for finetuning (Finn et al., 2017; Rajeswaran et al., 2019), and ones that do not (Snell et al., 2017; Lee et al., 2019). In the meta-learning phase, the meta-learner is trained to solve entire tasks as (meta-training) datapoints. Its generalization is measured by how well it can quickly adapt to novel test tasks. Many different strategies have been introduced to maximize the effectiveness of the meta-learning phase such as episodic training, where the model is trained by simulating 'test-like' conditions (Vinyals et al., 2016), or finetuning, where the model performs up to $m$ gradient steps on the new task (Finn et al., 2017).

While such meta-learning approaches assume the availability of a finetuning budget for adaptation at test time, zero-shot approaches introduce the limit scenario of fast adaptation, in which generalization has to be achieved without access to any examples. Such a setting can be found in metric learning (Yang & Jin, 2006; Suárez et al., 2018), where a model is evaluated on the ability to perform zero-shot retrieval on novel data. Most commonly, metric models are trained on a training data distribution $\mathcal{D}_{\text{train}}$ and evaluated on a testing distribution $\mathcal{D}_{\text{test}}$ which share no classes. However, the data generating function is assumed to be similar between $\mathcal{D}_{\text{train}}$ and $\mathcal{D}_{\text{test}}$, such as natural images of birds (Welinder et al., 2010). While vanilla metric learning learns a parametrized metric on fixed feature extractors, Deep Metric Learning (DML) leverages deep neural networks to train the feature extractors concurrently (Roth et al., 2020b). Such deep abstractions further allow for simplified and computationally cheap metrics such as euclidean distances which make large-scale retrieval applications with fast similarity searches possible (e.g. Johnson et al. (2017)).
Similar to DML, Zero-Shot Domain Adaptation (ZSDA) introduces a learner that is also trained and evaluated on two distinct $\mathcal{D}_{\text{train}}$ and $\mathcal{D}_{\text{test}}$ in a zero-shot setting. However, unlike DML, in ZSDA, the labels between the data distributions are shared. Instead, training and test distribution come from distinct data generative functions, such as natural images of digits (Goodfellow et al., 2013) and handwritten images of digits (LeCun, 1998).

## 3 EXTENDING NETWORK TRAINING WITH A UNIFORMITY PRIOR

In this section, we introduce the proposed *uniformity regularization* and detail the employed alternating GAN-like optimization scheme to perform it in a computationally tractable manner.

**Prior Matching.** Given a neural network $q(y|x)$ that is parameterized by $\theta$ we formally define the training objective as $\mathcal{L}_T(q(y|x), y)$ where $\mathcal{L}_T$ is any task-specific loss such as a cross-entropy loss, $(x, y)$ are samples from the training distribution $\mathcal{D}_{\text{train}}$ and $q(y|x)$ the probability of predicting label $y$ under $q$. This is a simplified formulation; in practice, there are many different ways to train a neural network, such as ranking-based training with tuples (Chopra et al., 2005). We define the embedding space $z$ as the output of the final convolutional layer of a deep network. Accordingly, we'll note $q(z|x)$ as the conditional distribution for that embedding space which, due to the convnet being a deterministic mapping, is a dirac delta distribution at the value of the final convolutional layer. Section 4.1 further details how to apply *uniformity regularization* in practice.

As we ultimately seek to impose a uniformity prior over the learned aggregate feature/embedding "posterior" $q(z) = \int_x q(z|x)p(x)dx$, we begin by augmenting the generic task-objective to allow for the placement of a prior $r(z)$. For priors $r(z)$ with closed-form KL-divergences $\mathbf{D}$, one can define a prior-regularized task objective as

$$\mathcal{L} = \min_{\theta} \mathbb{E}_{(x,y) \sim \mathcal{D}_{\text{train}}} \left[ \mathcal{L}_T(q(y|x), y) \right] + \mathbf{D}_{x \sim \mathcal{D}_{\text{train}}} \left( q(z|x) \| r(z) \right) \tag{3}$$

similar to the Variational Autoencoder formulation in Kingma & Welling (2013). However, to improve the generalization of a network by encouraging uniformity in the learned embeddings, we require regularization by matching the learned embedding space to a uniform distribution prior $\mathcal{U}(-\alpha, \beta)$, defined by the lower and upper bounds $\alpha$ and $\beta$, respectively. Unfortunately, such a regularization does not have a simple solution in practice, as a bounded uniform distribution has no closed-form KL divergence metric to minimize.

**Uniformity Regularization.** To address the practical limitation of solving Eqn. 3, we draw upon the GAN literature, in which alternate adversarial optimization has been successfully used to match

a generated distribution to a defined target distribution using implicit divergence minimization. Latent variable models such as the Adversarial Autoencoder (Makhzani et al., 2015) have successfully used such a GAN-style adversarial loss, instead of a KL divergence, in the latent space of the autoencoder to learn a rich posterior. Such implicit divergence minimization allows us to match any well-defined distribution as a prior, but more specifically, ensures that we can successfully match learned embedding spaces to $\mathcal{U}(-\alpha, \alpha)$, which we set to the unit hypercube $\mathcal{U}(-1, 1)$ by default.

To this end, we adapt the GAN objective in Eqn. 1 and 2 for uniformity regularization optimization and train a discriminator, $D$, to be an arbiter between which samples are from the learned distribution $q(z|x)$ and from the uniform prior $r(z)$. As such, the task model $q$ (parameterized by $\theta$) aims to *fool* the discriminator $D$ into thinking that learned features, $q(z|x)$, come from the chosen uniform target distribution, $r(z)$, while the discriminator $D$ learns to distinguish between learned features and samples taken from the prior, $\tilde{z} \sim r(z)$. Note that while the task-model defines a deterministic mapping for $q(z|x)$ instead of a stochastic one, the aggregate feature "posterior" $\int_x q(z|x)p(x)dx$, on which we apply our uniformity prior, is indeed a stochastic distribution (Makhzani et al., 2015).

Concretely for our *uniformity regularization*, we rewrite the discriminator objective from Eqn. 1 to account for the uniform prior matching, giving

$$\mathcal{L}_D = \max_D \mathbb{E}_{x \sim \mathcal{D}_{\text{train}}}[\log(1 - D(q(z|x)))] + \mathbb{E}_{\tilde{z} \sim \mathcal{U}(-1,1)}[\log D(\tilde{z})] \tag{4}$$

Consequently, we reformulate the generator objective from Eqn. 2 to reflect the task-model $q$,

$$\mathcal{L}_{\text{max}} = \min_\theta \mathbb{E}_{x \sim \mathcal{D}_{\text{train}}}[\log(1 - D(q(z|x)))] \tag{5}$$

where we used the notation $\mathcal{L}_{\text{max}}$ to reflect that optimization maximizes the feature uniformity by learning to fool $D$. Our final min-max *uniformity regularized* objective for $\theta$ and the Discriminator is then given as

$$\mathcal{L} = \min_\theta \max_D \mathbb{E}_{(x,y) \sim \mathcal{D}_{\text{train}}}[\mathcal{L}_T(q_\theta(y|x), y)] + \gamma \mathbb{E}_{x \sim \mathcal{D}_{\text{train}}}[\log(1 - D(q_\theta(z|x)))] + \mathbb{E}_{\tilde{z} \sim \mathcal{U}(-1,1)}[\log D(\tilde{z})] \tag{6}$$

with task-objective $\mathcal{L}_T$ and training data distribution $\mathcal{D}_{\text{train}}$. Using this objective, the learned feature space is implicitly encouraged to become more uniform. The amount of regularization is controlled by the hyperparameter $\gamma$, balancing generalization of the model to new tasks and performance on the training task at hand. Large $\gamma$ values hinder effective feature learning from training data, while values of $\gamma$ too small result in weak regularization, leading to a non-uniform learned feature distribution with reduced generalization capabilities.

## 4 EXPERIMENTS

We begin by highlighting the link between feature space uniformity and generalization performance (§4.2). In a large-scale experimental study covering settings in which samples are available for adaptation (Meta-Learning, §4.3), or not (Deep Metric Learning & Zero-Shot Domain Adaptation, §4.4) and Out-of-Distribution Detection (§4.5), we then experimentally showcase how *uniformity regularization* can facilitate generalizability of learned features and the ability of a model to perform fast adaptation to novel tasks and data. For all experiments, *we do not perform hyperparameter tuning on the base algorithms*, and use the same hyperparameters that the respective original papers proposed; we simply add the *uniformity regularization*, along with the task loss as in Eqn. 6.

### 4.1 EXPERIMENTAL DETAILS

*Uniformity regularization* was added to the output of the CNNs for all networks. For ResNet-variants (He et al., 2016; Xie et al., 2017; Zagoruyko & Komodakis, 2016), it was applied to the output of the CNNs, just before the single fully-connected layer. For meta-learning, the regularization is applied directly on the learned metric space for the metric-space based meta-learners (Vinyals et al., 2016; Snell et al., 2017; Liu et al., 2020), and applied to the output of the penultimate layer for MAML (Finn et al., 2017). The discriminator is parameterized using a three-layer MLP with 100 hidden units in each layer and trained using the Adam optimizer (Kingma & Ba, 2014) with a learning rate of $10^{-5}$. The value of $\gamma$ is chosen to be 0.1 for all experiments, except for Deep Metric Learning. For Deep Metric Learning, a value of $\gamma = 0.4$ is chosen, since the effect of regularization needs

Table 1: **Influence of Feature Space Uniformity on Generalization.** We study the influence of feature space uniformity on the generalization capabilities in ZSDA by matching the feature space to prior distributions $r(z)$ of increasing uniformity (left to right). We report mean accuracy and standard deviation over 5 runs on the task of MNIST $\rightarrow$ USPS and USPS $\rightarrow$ MNIST zero-shot domain adaptation using ResNet-18.

| Task | Baseline | $\mathcal{N}(0, 0.1 \times \mathcal{I})$ | $\mathcal{N}(0, \mathcal{I})$ | $\mathcal{N}(0, 5 \times \mathcal{I})$ | $\mathcal{N}(0, 10 \times \mathcal{I})$ | $\mathcal{U}(-1, 1)$ |
|---|---|---|---|---|---|---|
| MNIST $\rightarrow$ USPS | $49.0 \pm 0.20$ | $43.98 \pm 0.23$ | $43.45 \pm 0.16$ | $56.45 \pm 0.36$ | $59.80 \pm 0.12$ | $\mathbf{67.2} \pm 0.11$ |
| USPS $\rightarrow$ MNIST | $42.8 \pm 0.07$ | $27.23 \pm 0.28$ | $26.02 \pm 0.87$ | $37.96 \pm 0.32$ | $43.76 \pm 0.48$ | $\mathbf{56.2} \pm 0.10$ |

to be stronger, as Deep Metric Learners (commonly a ResNet-50 (He et al., 2016) or Inception-V1 (Szegedy et al., 2016) with Batch-Norm (Ioffe & Szegedy, 2015)) start off with networks that are already pre-trained on ImageNet (Russakovsky et al., 2015).

## 4.2 FEATURE SPACE UNIFORMITY IS LINKED TO GENERALIZATION PERFORMANCE

We first investigate the connection between feature space uniformity and generalization, measured by generalization performance in Zero-Shot Domain Adaptation (more experimental details in §4.4). Unfortunately, the uniform hypercube prior in our *uniformity regularizer* does not provide a way for intuitive and explicit uniformity scaling - one can not make the uniform prior "more or less uniform". As such, we make use of a Gaussian prior $\mathcal{N}(\mu, \sigma^2)$. Under the fair assumption that the learned embedding space of deep neural networks does not have infinite support in practice (especially given regularization methods such as L2 regularization), the variance $\sigma^2$ provides a uniformity scaling factor - with increased variance, the Gaussian prior reduces mass placed around embeddings near $\mu$, effectively encourageing the network to learn a more uniform embedding space. We can therefore directly evaluate the importance of feature space uniformity by using our GAN-based regularization scheme to match feature space distribution to Gaussian priors with different $\sigma^2$ scales.

Using this setup, Table 1 compares feature space uniformity against the model's ability to perform ZSDA from MNIST to USPS (and respective backward direction) using a ResNet-18 (He et al., 2016). As can be seen, when the uniformity of the (Gaussian) prior $r(z)$ is increased, the ability to perform domain adaptation also improves. When $\sigma^2$ is small, the model is unable to effectively adapt to the novel data, and as the uniformity of $r(z)$ is increased, the network significantly improves its ability to perform the adaptation task. We also find that maximal performance is achieved at maximum uniformity, which corresponds to our uniform hyper-cube prior $\mathcal{U}(-1, 1)$ and coincides with insights made in Wang & Isola (2020) and Roth et al. (2020b). The impact of our *uniformity regularization* is even more evident on the backward task of USPS $\rightarrow$ MNIST, since there are less labels present in the USPS dataset, thereby making overfitting a greater issue when trained on USPS.

## 4.3 UNIFORM PRIORS BENEFIT META-LEARNING

We now study the influence of *uniformity regularization* on meta-training for few-shot learning tasks, which we divide into two experiments. First, we evaluate how *uniformity regularization* impacts the performance of three distinct meta-learning baselines: Matching Networks (Vinyals et al., 2016), Prototypical Networks (Snell et al., 2017) and MAML (Finn et al., 2017). Performance is evaluated on four few-shot learning benchmarks: Double MNIST (LeCun, 1998), Omniglot (Lake et al., 2019), CIFAR-FS (Krizhevsky et al., 2009) and MiniImagenet (Vinyals et al., 2016). For our implementation, we utilize TorchMeta (Deleu et al., 2019). Results for each meta-learning method with and without regularization are summarized in Table 2a)[2]. As can be seen, the addition of *uniformity regularization* benefits generalization across method and benchmark, in some cases notably. This holds regardless of the number of shots used at meta-test-time, though we find the largest performance gains in the 1-shot scenario. In addition, when compared to other regularization methods such as Dropout (Srivastava et al., 2014), L2-regularization (Tibshirani, 1996) and hyper-spherical uniformity regularization Wang & Isola (2020), it compares favorably, especially on more complex datasets such as MiniImageNet. Compared to Wang & Isola (2020), this is especially impressive given the much wider application range. Overall, the results highlight the benefit of reduced training-task bias introduced by *uniformity regularization* for fast adaptation to novel test tasks.

---

[2]For Double MNIST and Omniglot, error rates are listed instead of accuracies.

Table 2: **Meta-Learning**. **1)** Comparison of several meta-learning algorithms on four few-shot learning benchmarks: Omniglot (Lake et al., 2019), Double MNIST (LeCun, 1998), CIFAR-FS (Krizhevsky et al., 2009) and Mini-Imagenet Vinyals et al. (2016). We test with multiple regularization techniques such as Dropout, L2 regularization and compare directly against uniformity-alignment (U-A) as proposed by Wang & Isola (2020). The models are evaluated with and without *uniformity regularization* ($\mathcal{UR}$) and we report the mean **error rate** over 5 seeds. No hyperparameter tuning is performed on the meta-learner and we use the exact hyperparameters as proposed in the original paper. **2)** Application of *uniformity regularization* with Universal Representation Transformer Layers (Liu et al., 2020) on Meta-Dataset to improve further upon the state-of-the-art performance of URT. Numbers listed in **blue** represent the current state-of-the-art on the MetaDataset tasks.

| 1) Baseline Study | Omniglot | | Double MNIST | | CIFAR-FS | | MiniImageNet | |
|---|---|---|---|---|---|---|---|---|
| Methods ↓ | (5, 1) | (5,5) | (5, 1) | (5,5) | (5, 1) | (5,5) | (5, 1) | (5,5) |
| MAML | **4.8**± 0.4 | **1.5** ± 0.4 | **7.9**± 0.7 | **1.9**± 0.3 | **52.1** ± 0.8 | **67.1** ± 0.9 | 47.2 ± 0.7 | **62.1** ± 1.0 |
| MAML + $\mathcal{UR}$ | **4.1**± 0.5 | **1.3**± 0.2 | **7.3** ± 0.2 | **1.5** ± 0.5 | **52.9** ± 0.4 | **67.1** ± 0.9 | **48.9** ± 0.8 | **64.1** ± 1.0 |
| Matching Networks | 2.1 ± 0.2 | **1.0** ± 0.2 | 4.2 ± 0.2 | **2.7** ± 0.2 | 46.7 ± 1.1 | **62.9** ± 1.0 | 43.2 ± 0.3 | 50.3 ± 0.9 |
| Matching Networks + Dropout | 2.4 ± 0.2 | 1.3 ± 0.2 | 4.4 ± 0.2 | 2.9 ± 0.4 | 45.3 ± 1.1 | **63.0** ± 0.7 | 42.9 ± 0.9 | 50.0 ± 1.0 |
| Matching Networks + L2 reg. | 2.1 ± 0.2 | **1.0** ± 0.1 | 4.1 ± 0.2 | 2.6 ± 0.2 | 46.9 ± 1.1 | **63.0** ± 0.9 | 43.3 ± 0.8 | 50.1 ± 1.0 |
| Matching Networks + U-A | 2.0 ± 0.1 | **0.9** ± 0.1 | 3.9 ± 0.3 | **2.7** ± 0.1 | 47.3 ± 1.0 | **63.1** ± 0.8 | 43.5 ± 0.7 | 50.3 ± 1.0 |
| Matching Networks + $\mathcal{UR}$ | **1.7**± 0.1 | **0.9**± 0.1 | **3.2**± 0.1 | **2.3**± 0.3 | **49.3** ± 0.4 | **63.1** ± 0.7 | **47.1** ± 0.8 | **53.1** ± 0.7 |
| Prototypical Network | **1.6** ± 0.2 | **0.4** ± 0.1 | **1.3** ± 0.2 | **0.2** ± 0.2 | 52.4 ± 0.7 | 67.1 ± 0.5 | 45.4 ± 0.6 | 61.3 ± 0.7 |
| Prototypical Network + Dropout | 1.9 ± 0.2 | **0.5** ± 0.2 | **1.4** ± 0.2 | 0.5 ± 0.1 | 51.9 ± 0.8 | **66.0** ± 0.4 | 44.8 ± 0.7 | 61.2 ± 0.9 |
| Prototypical Network + L2 reg. | **1.6** ± 0.2 | **0.4** ± 0.1 | **1.3** ± 0.1 | 0.3 ± 0.2 | **52.5** ± 0.8 | 66.3 ± 0.4 | 45.0 ± 0.7 | 61.4 ± 0.7 |
| Prototypical Network + U-A | **1.5** ± 0.3 | **0.4** ± 0.1 | **1.2** ± 0.1 | **0.2** ± 0.2 | 52.6 ± 0.7 | 66.3 ± 0.5 | 45.4 ± 0.5 | 61.8 ± 0.8 |
| Prototypical Network + $\mathcal{UR}$ | **1.2** ± 0.3 | **0.4** ± 0.1 | **1.0** ± 0.2 | **0.2** ± 0.2 | 52.6 ± 0.8 | **66.8** ± 0.5 | **46.8** ± 0.5 | **64.4** ± 0.9 |

| 2) Meta-Dataset | ILSVRC | Omniglot | Aircrafts | Birds | Textures | QuickDraw | Fungi | VGGFlower | TrafficSigns | MSCOCO | Avg. Rank |
|---|---|---|---|---|---|---|---|---|---|---|---|
| TaskNorm | 50.6 ± 1.1 | 90.7 ± 0.6 | 83.8 ± 0.6 | 74.6 ± 0.8 | 62.1 ± 0.7 | 74.8 ± 0.7 | 48.7 ± 1.0 | 89.6 ± 0.6 | 67.0 ± 0.7 | 43.4 ± 1.0 | 4.5 |
| SUR | 56.3 ± 1.1 | 93.1 ± 0.5 | 85.4 ± 0.7 | 71.4 ± 1.0 | 71.5 ± 0.8 | 81.3 ± 0.8 | **63.1** ± 1.0 | 82.8 ± 0.7 | 70.4 ± 0.8 | 52.4 ± 1.1 | 3.2 |
| SimpleCNAPS | **58.6** ± 1.1 | 91.7 ± 0.6 | 82.4 ± 0.7 | 74.9 ± 0.8 | 67.8 ± 0.8 | 77.7 ± 0.7 | 46.9 ± 1.0 | **90.7** ± 0.5 | **73.5** ± 0.7 | 46.2 ± 1.1 | 3.2 |
| URT | 55.7 ± 1.0 | 94.4 ± 0.4 | 85.8 ± 0.6 | **76.3** ± 0.8 | 71.8 ± 0.7 | 82.5 ± 0.6 | **63.5** ± 1.0 | 88.2 ± 0.6 | 69.4 ± 0.8 | 52.2 ± 1.1 | 2.6 |
| URT + $\mathcal{UR}$ | **58.3** ± 0.9 | **95.2** ± 0.2 | **88.0** ± 0.9 | **76.7** ± 0.8 | **74.9** ± 0.9 | **84.0** ± 0.3 | 62.8 ± 1.1 | **90.3** ± 0.4 | **72.9** ± 0.8 | **54.6** ± 1.1 | **1.5** |

To measure the benefits for large-scale few-shot learning problems, we further examine *uniformity regularization* on the Meta-Dataset Triantafillou et al. (2019), which contains data from diverse domains such as natural images, objects and drawn characters. We follow the setup suggested by Triantafillou et al. (2019), used in Liu et al. (2020), in which eight out of the ten available datasets are used for training, while evaluation is done over all. Results are averaged across varying numbers of ways and shots. We apply *uniformity regularization* on the state-of-the-art Universal Representation Transformer (URT) (Liu et al., 2020), following their implementation and setup without hyperparameter tuning. Table 2b), *uniformity regularization* shows consistent improvements upon URT, matching or even outperforming the state-of-the-art on all sub-datasets.

## 4.4 Uniform Priors benefit Zero-Shot Generalization

Going further, we study limit cases of fast adaption and look at how *uniformity regularization* affects zero-shot retrieval in Deep Metric Learning and zero-shot classification for domain adaptation. Here, the model is evaluated on a different distribution than the training distribution without finetuning, highlighting the benefits of *uniformity regularization* for learning task-agnostic & reusable features.

**Deep Metric Learning.** We apply *uniformity regularization* on four benchmark DML objectives (Contrastive Loss (Hadsell et al., 2006), Margin Loss (Wu et al., 2017), Softmax Loss (Zhai & Wu, 2018) and MultiSimilarity Loss (Wang et al., 2019)) studied in Roth et al. (2020b), and evaluate them over two standard datasets: CUB-200 (Welinder et al., 2010), and Cars-196 (Krause et al., 2013). The results summarized in Table 3a) reveal substantial gains in performance measured over all evaluation metrics across all benchmarks and a diverse set of baselines. Less competitive methods such as Zhai & Wu (2018) are even able significantly outperform all non-regularized baselines on CUB-200 (Welinder et al., 2010) when regularized. This showcases the effectiveness in fighting against overfitting for generalization, even without finetuning at test-time.
Finally, when evaluated in two different common literature settings and compared against, in parts much more complex, state-of-the-art methods, we find that simple *uniformity regularized* objectives can match or even outperform these, in some cases significantly.

Table 3: **Deep Metric Learning (Zero-Shot Generalization)**. **1)** Evaluation of *uniformity regularization* ($\mathcal{UR}$) on strong deep metric learning baseline objectives with a ResNet-50 backbone (He et al., 2016) on two standard benchmarks: CUB200-2011 (Welinder et al., 2010) & CARS196 (Krause et al., 2013). We report mean Recall@1 and Normalized Mutual Information (NMI). All baseline scores are taken from Roth et al. (2020b), and their official released code is used to run all experiments without any fine-tuning. **2)** We show that with standard learning rate scheduling, *uniformity regularization* ($\mathcal{UR}$) can provide strong performance boosts on baseline objectives to reach and even improve upon state-of-the-art methods in DML, measured on two standard setups with a ResNet-50 backbone and an Inception-V1 network with frozen Batch-Normalization. Numbers listed in **blue** represent the current state-of-the-art on the benchmark dataset over the given metric.

| 1) Ablation Study | | | | CUB200-2011 | | CARS196 | |
|---|---|---|---|---|---|---|---|
| Methods ↓ | ‖ | Backbone | Embed. Dim. ‖ | R@1 | NMI ‖ | R@1 | NMI |
| Softmax (Zhai & Wu, 2018) | ‖ | ResNet-50 | 128 | $61.7 \pm 0.3$ | $66.8 \pm 0.4$ | $78.9 \pm 0.3$ | $66.4 \pm 0.3$ |
| Softmax + $\mathcal{UR}$ | ‖ | ResNet-50 | 128 | $\mathbf{65.0 \pm 0.1}$ | $\mathbf{68.8 \pm 0.2}$ | $\mathbf{80.6 \pm 0.2}$ | $\mathbf{68.3 \pm 0.2}$ |
| Margin (D, 1.2) (Wu et al., 2017) | ‖ | ResNet-50 | 128 | $63.1 \pm 0.5$ | $68.2 \pm 0.3$ | $79.9 \pm 0.3$ | $67.4 \pm 0.3$ |
| Margin (D, 1.2) + $\mathcal{UR}$ | ‖ | ResNet-50 | 128 | $\mathbf{65.0 \pm 0.3}$ | $\mathbf{69.5 \pm 0.2}$ | $\mathbf{82.5 \pm 0.1}$ | $\mathbf{68.9 \pm 0.2}$ |
| Multisimilarity (Wang et al., 2019) | ‖ | ResNet-50 | 128 | $62.8 \pm 0.7$ | $68.6 \pm 0.4$ | $81.7 \pm 0.2$ | $69.4 \pm 0.4$ |
| Multisimilarity + $\mathcal{UR}$ | ‖ | ResNet-50 | 128 | $\mathbf{65.4 \pm 0.4}$ | $\mathbf{70.3 \pm 0.3}$ | $\mathbf{82.2 \pm 0.2}$ | $\mathbf{70.5 \pm 0.3}$ |
| **2) Literature Comparison** | | | | CUB200-2011 | | CARS196 | |
| Methods ↓ | ‖ | Backbone | Embed. Dim. ‖ | R@1 | NMI | R@1 | NMI |
| Div&Conq (Sanakoyeu et al., 2019) | ‖ | ResNet-50 | 128 | 65.9 | 69.6 | **84.6** | 70.3 |
| MIC (Roth et al., 2019) | ‖ | ResNet-50 | 128 | 66.1 | 69.7 | 82.6 | 68.4 |
| PADS (Roth et al., 2020a) | ‖ | ResNet-50 | 128 | **67.3** | 69.9 | 83.5 | 68.8 |
| Multisimilarity+$\mathcal{UR}$ | ‖ | ResNet-50 | 128 | $66.3 \pm 0.4$ | $\mathbf{70.5 \pm 0.3}$ | $84.0 \pm 0.2$ | $\mathbf{71.3 \pm 0.5}$ |
| Multisimilarity (Wang et al., 2019) | ‖ | Inception-V1 + BN | 512 | 65.7 | - | 84.1 | - |
| Softtriple (Qian et al., 2019) | ‖ | Inception-V1 + BN | 512 | 65.4 | 69.3 | 84.5 | 70.1 |
| Group (Elezi et al., 2019) | ‖ | Inception-V1 + BN | 512 | 65.5 | 69.0 | **85.6** | **72.7** |
| Multisimilarity+$\mathcal{UR}$ | ‖ | Inception-V1 + BN | 512 | $\mathbf{68.5 \pm 0.3}$ | $\mathbf{71.7 \pm 0.5}$ | $\mathbf{85.8 \pm 0.3}$ | $\mathbf{72.2 \pm 0.5}$ |

**Zero-Shot Domain Adaptation.** For Zero-Shot Domain Adaptation, we conduct digit recognition experiments, transferring models between MNIST (LeCun, 1998), SVHN (Goodfellow et al., 2013) and USPS (Seewald, 2005). In this setting, we train the model on a source dataset, and test it directly on the test dataset. Since each of the datasets contain digits, the networks are assessed on their ability to classify digits on the target dataset, without any training. We evaluate different architectures, LeNet (LeCun et al., 1998) and ResNet-18 (He et al., 2016), as well as a distinct domain adaptation approach (Adversarial Discriminative Domain Adaptation, ADDA) (Tzeng et al., 2017)).

Results in Tab. 4 show that when training on only the source data, networks with *uniformity regularization* significantly outperform baseline models by as much as $18\%$ on the target dataset. The gain in performance for ResNets and LeNets trained only on the source data demonstrates that such models disproportionately overfit to the training (or source) data, which we can alleviate via *uniformity regularization* to learn better data-agnostic features. Performance gains are also evident in ADDA, which operates under an adversarial training setting different from "Source Only" baseline models. In addition, ADDA with *uniformity regularization* achieves Zero-Shot Domain Adaptation performance close to that of a supervised learner trained directly on target data ("Target Only").

These improvements over two distinct model architectures and ADDA further showcase the generality of the proposed regularization technique in learning better fast-adaptive models.

## 4.5 UNIFORM PRIORS BENEFIT OUT-OF-DISTRIBUTION GENERALIZATION

In this section, we evaluate trained models on their benefits to the detection of Out-of-Distribution (OOD) data. We perform severe image augmentations using random translations of $[-4, 4]$ pixels, random rotations between $[-30, 30]$ degrees and scaling by a factor between $[0.75, 1.25]$. These transformations are physical transformations to an image, and completely preserve the semantics of the image. We evaluate three state-of-the-art architectures (He et al., 2016; Xie et al., 2017; Zagoruyko & Komodakis, 2016) on their classification generalization performance to OOD CIFAR-10 data (Krizhevsky et al., 2009). The improvements in classification accuracies again show the disproportionate usefulness of *uniformity regularization* (table 5). Similar to domain adaptation, we find that the network trained on a defined downstream task is able to significantly improve its ability to generalize to a new data distribution when *uniformity regularization* is added. We note

Table 4: **Zero-Shot Domain Adaptation**. Comparison of several zero-shot domain adaptation strategies on the digit recognition task. The models are evaluated with and without *uniformity regularization* ($\mathcal{UR}$) and we report the mean accuracy and standard deviation over 5 random seeds. The results for Adversarial Domain Discriminative Adaptation (ADDA) and the "Source Only" + LeNet backbone are taken directly from Tzeng et al. (2017). "Target Only" refers to a model directly being trained and evaluated on the target distribution. We perform no hyperparameter tuning, and the exact hyperparameters are used as in Tzeng et al. (2017).

| Source → Target | Backbone | MNIST → USPS | USPS → MNIST | SVHN → MNIST |
|---|---|---|---|---|
| Source Only | ResNet-18 | $49.0 \pm 0.20$ | $42.8 \pm 0.07$ | $69.7 \pm 0.06$ |
| Source Only + $\mathcal{UR}$ | ResNet-18 | $\mathbf{67.2} \pm 0.11$ | $\mathbf{56.2} \pm 0.10$ | $\mathbf{71.3} \pm 0.13$ |
| Source Only | LeNet | $75.2 \pm 0.016$ | $57.1 \pm 0.017$ | $60.1 \pm 0.011$ |
| Source Only + $\mathcal{UR}$ | LeNet | $\mathbf{79.6} \pm 0.04$ | $\mathbf{62.6} \pm 0.01$ | $\mathbf{65.8} \pm 0.03$ |
| ADDA (Tzeng et al., 2017) | LeNet | $89.4 \pm 0.002$ | $90.1 \pm 0.008$ | $76.0 \pm 0.018$ |
| ADDA + $\mathcal{UR}$ | LeNet | $\mathbf{93.5} \pm 0.09$ | $\mathbf{94.8} \pm 0.03$ | $\mathbf{81.6} \pm 0.03$ |
| Target Only | ResNet-18 | $98.1 \pm 0.2$ | $99.8 \pm 0.1$ | $99.8 \pm 0.1$ |

Table 5: **Out-of-Distribution Generalization**. Comparison of OOD detection performance for various networks with mean accuracy and standard deviation over 5 seeds. To generate OOD samples, we perform random translations, rotations, and scaling over each test image. We use $\mathcal{UR}$ to denote *uniformity regularization*.

| ResNet-18 | + $\mathcal{UR}$ | WideResNet-50 | + $\mathcal{UR}$ | ResNeXt-50 | + $\mathcal{UR}$ |
|---|---|---|---|---|---|
| $35.6 \pm 1.2$ | $\mathbf{41.3} \pm 1.3$ | $39.6 \pm 1.2$ | $\mathbf{43.9} \pm 0.9$ | $40.1 \pm 0.8$ | $\mathbf{43.8} \pm 1.1$ |

that the reason for *relatively* high variance in the results is due to the stochastic nature of the data augmentation techniques. The data augmentation during evaluation randomly performs a physical transformation on the image, which means that the different models will likely see unique augmentations of the same image during evaluation, which can explain the high standard deviation in results.

### 4.6 How Uniform Priors change the feature space

Finally, we examine both qualitatively and quantitatively the influence of *uniformity regularization* on the feature space, for which we look at the feature space changes in the Deep Metric Learning problem. DML naturally lends itself to this study, as changes in the feature space are more strongly reflected in the underlying objective, which is to learn generalizing embedding spaces operating on top of these features. Figure 1 qualitatively shows increased feature space density[3] when applying *uniformity regularization* and mapping to 2-d with UMAP McInnes et al. (2020), especially on the test data, showcasing that *uniformity regularization* has an impact on the feature distribution. As important and subtle changes are likely lost in the dimensionality reduction process, we also perform a quantitative evaluation of the actual feature space density, following the definition in Roth et al. (2020b). Here, we find a 30% increase on the training feature density when applying *uniformity regularization*. This is consistent with the observations made in Roth et al. (2020b), which link increased embedding space density on the training data to improved generalization to the test data[4].

## 5 Related Work

**Adversarial Representation Learning.** Latent variable models (e.g. Makhzani et al. (2015); Tolstikhin et al. (2017)) have used GAN-style training in the latent space to learn a rich posterior. Recent efforts have made such training effective in different contexts like active learning (Sinha et al., 2019; Kim et al., 2020) or domain adaptation (Tzeng et al., 2017; Hoffman et al., 2017). Our work more closely follows Hjelm et al. (2019), who applied a similar adversarial objective to impose certain properties on the features learned through self-supervised representation learning. While Hjelm et al. (2019) also matched their representations to a uniform distribution, no detailed

---

[3]Ratio of sample distances within shared classes and distances between class clusters (Roth et al., 2020b).

[4]We note that Roth et al. (2020b) performed their embedding space studies on final embeddings produced by a linear mapping from the feature space. Thus, we believe insights to be transferable.

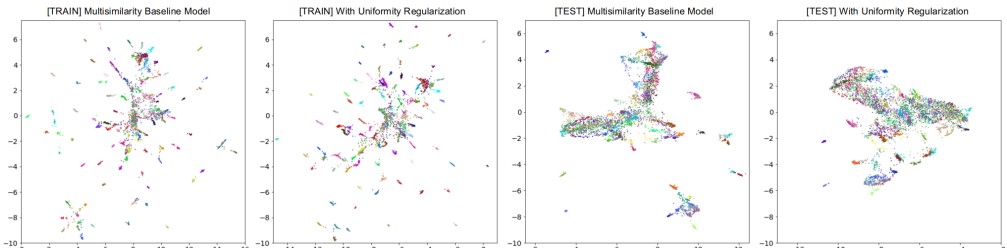

Figure 1: **Qualitative feature space study.** Evaluation of feature space changes caused by imposing a uniformity prior and producing a 2-d feature map using UMAP McInnes et al. (2020). The left two plots are the baseline and baseline with uniformity distribution on the training data; the right two plots are the baseline and baseline with uniformity distribution on the testing data. We see that visually, the density increases with uniformity regularization, especially notable on the test data. Such embedding space density improvement can be linked to generalization Roth et al. (2020b), which we also quantitatively support in section 4.6.

reasons for this specific choice were given. Instead, our work shows that it is exactly the uniformity of feature distributions introduced by our *uniformity regularization* that facilitate fast adaptation and transfer to novel data and tasks in neural networks, regardless of the specific application domain.

**Deep Metric Learning and Generalization.** The goal of a Deep Metric Learning (DML) algorithm is to learn a metric space that encodes semantic relations as distances and which generalizes sufficiently that at test-time zero-shot retrieval on novel classes and samples can be performed. Representative methods in DML commonly differ in their proposed objectives (Wu et al., 2017; Wang et al., 2019; Zhai & Wu, 2018; Hadsell et al., 2006; Chen et al., 2017), which are commonly accompanied with tuple sample methods (Wu et al., 2017; Harwood et al., 2017; Wang et al., 2016; Roth et al., 2020a). Extension to the basic training paradigm, such as with self-supervision (Milbich et al., 2020; Cao et al., 2019) have also shown great promise. Recently, Roth et al. (2020b) have performed an extensive survey on the various DML objectives to study driving factors for generalization among these methods. In that regard, recent work by Wang & Isola (2020) has offered theoretical insights into the benefits of learning on a Uniform hypersphere for zero-shot generalization.
Achieving Out-of-Distribution generalization from different point of views has also been of great interest, ranging from work on zero-shot domain adaptation (Kodirov et al., 2015; Tzeng et al., 2017; Hoffman et al., 2017) to the study of invariant correlations (Arjovsky et al., 2019).

**Meta-Learning** Many types of meta-learning algorithms for few-shot learning (but also for zero-shot learning such as Pal & Balasubramanian (2019)) have recently been proposed, building on memory-augmented methods (Ravi & Larochelle, 2016; Munkhdalai et al., 2017; Santoro et al., 2016), metric-based approaches (Vinyals et al., 2016; Snell et al., 2017; Sung et al., 2018) or optimization-based techniques (Lee et al., 2019; Finn et al., 2017; Rajeswaran et al., 2019; Yin et al., 2019). Finetuning using ImageNet pretraining (Chen et al., 2020; Gidaris & Komodakis, 2018) has also been proposed as alternative approaches. Meta-learning has also been explored for fast adaptation of novel tasks in reinforcement learning (Kirsch et al., 2019; Zintgraf et al., 2019; Jabri et al., 2019). More closely related to our approach is Jamal et al. (2018), which propose to use inequality measures between different tasks to learn less task-dependent representations. However, this approach is still limited to episodic learning akin to most few-shot learning approaches. *Uniformity regularization* is much more generic, being applicable to domains outside of Meta-Learning, and does not dependent on the choice of inequality measure.

## CONCLUSION

In this paper, we propose a regularization technique for the challenging task of fast adaptation to novel tasks. We present a simple and general solution, *uniformity regularization*, to reduce training bias and encourage networks to learn more reusable features. In a large experimental study, we show benefits across multiple, distinct domains studying varying degrees of fast adaptation and generalization such as Meta-Learning, Deep Metric Learning, Zero-Shot Domain Adaptation and Out-of-Distribution Detection, and highlight the role of uniformity of the prior over learned features for generalization and adaptation. For Meta-Learning and Deep Metric Learning, we further show that simple *uniformity regularization* can offer state-of-the-art performance.

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
