# OpenReview forum: "Uniform Priors for Data-Efficient Transfer"
_ICLR.cc/2021/Conference — Reject_

### Official Review · AnonReviewer2 · 2020-10-28
**Review of AnonReviewer2**

**Rating:** 6
**Confidence:** 4

**Review:**

Summary:This paper broadly discusses about learning good transferable features in deep networks to perform few or zero-shot
adaptation to novel tasks. The paper proposes a uniformity regularization scheme that encourages better transfer and feature reuse. The method is validated on several benchmark datasets.

+ves:
+ The paper is overall well-written and easy to follow.
+ The paper has shown experimentations on different datasets under different learning setups, including deep metric learning and zero shot domain adaptation.
+ The overall idea of using a uniform regularizer on manifold is interesting and supported by theoretical considerations coming from GAN and VAE-like formulations.

Concerns:
- Equation 6, which is the final proposed loss, seems incomplete, which seems like a key issue. As I understand, Eqn 6 must consider the discriminator loss described in Eqn 4 too. Considering only the generator (that is the network q(z|x) till the last layer) is a necessary condition but may not be a sufficient condition to optimize the end-to-end network. I will be happy to know if I have missed something here.
- The paper seems to have missed some key references that have addressed similar problems - [1] and [2] listed at the end of this review. Both of these efforts show task transferability for computer vision tasks. This is important, considering the similarity of objectives.
- The paper states that it performs transfer between MNIST, SVHN and USPS (model trained on source data and tested on target data). It was not clear how MNIST --> SVHN was done. Wouldn't there be a mismatch of input channels in this case (SVHN is color, while MNIST is not).
- A feature space visualization of the proposed method and other baselines would have been very useful to directly compare and understand the claim and advantages of imposing uniform regularizer.
- One broader question that may be relevant (may not directly relate to a decision on the paper): How will uniformity on task space be affected if we leveraged a method such as Mixup [3]? How will the task space, in that case, look like?

Minor comments:
1.  Abstract: learn-ing --> learning,  transfer-able -->  transferable
2. Please check the line "... for improved generalization: For Unsupervised Representation Learning"
3. Section Uniformity Regularization: isn't \U(-\aplha, \beta) a correct formulation considering the previous \U(-\alpha, \beta) in section Prior Matching?
4. Page 7 Zero shot Domain Adaption: Please remove ``(" from the ``(LeNet''

References:
[1] Zero-shot task transfer, Pal et al, CVPR 2019
[2] LSM: Learning Subspace Minimization for Low-level Vision, Tang et al, CVPR 2020
[3] mixup: BEYOND EMPIRICAL RISK MINIMIZATION, Zhang et al, ICLR 2018

POST-REBUTTAL:
I thank the authors for their response.

At the outset, I am satisfied with the authors' responses to my questions - all the questions were answered. I do agree with other reviewers that the idea is incremental. Learning the prior across tasks is not very novel, as pointed out in references cited by other reviewers. However, on the bright side, the authors have done a good job in answering the questions, and the comprehensive experimental results are promising. The overall idea looks a bit incremental from the GAN literature side but maybe a good lead for meta and incremental learning literature.

I change my decision to "Weak Accept".

---

> ### Author Response · Authors · 2020-11-19
> **Response to AnonReviewer2**
>
> Thank you for your review, we have addressed each point individually.
>
> __1)  “Equation 6, which is the final proposed loss, seems incomplete, which seems like a key issue. As I understand, Eqn 6 must consider the discriminator loss described in Eqn 4 too. Considering only the generator (that is the network q(z|x) till the last layer) is a necessary condition but may not be a sufficient condition to optimize the end-to-end network. I will be happy to know if I have missed something here.”__
>
> Thank you for the correction! We have fixed Equation 6 where we have written the full min-max objective of the task-network and Discriminator.
>
>
> __2)  “The paper seems to have missed some key references that have addressed similar problems - [1] and [2] listed at the end of this review. Both of these efforts show task transferability for computer vision tasks. This is important, considering the similarity of objectives.”__
>
> Thank you for the reference! We have added the references in the related work. We do note that our method is strictly different from the mentioned works, with the main commonality being the study of generalization, however in different setups.
> We have experimentally shown that our method can work synergistically with many other methods and across diverse experiments.
>
>
> __3) “ The paper states that it performs transfer between MNIST, SVHN and USPS (model trained on source data and tested on target data). It was not clear how MNIST --> SVHN was done. Wouldn't there be a mismatch of input channels in this case (SVHN is color, while MNIST is not).”__
>
> This is a common task in zero-shot domain adaptation (Tzeng et al., Table 2). To perform experiments with SVHN -> MNIST, we simply repeat the single channel of the MNIST images 3-times for testing.
> Tzeng et al., 2017 Adversarial Discriminative Domain Adaptation, CVPR 2017
>
> __4)  “A feature space visualization of the proposed method and other baselines would have been very useful to directly compare and understand the claim and advantages of imposing uniform regularizer.”__
>
> Thank you for the suggestion. We have included UMAPs of the features on the training and test data for our deep metric learning study. We see that by adding uniformity regularization, improvements on the density of the embeddings can be seen, especially notable on the test data.
> This shows that uniformity regularization does indeed have an influence on the feature space.
> To further quantify this influence, we have also quantitatively evaluated the embedding space density following Roth et al. over the non-reduced features, as notable context was likely lost during the process of significant dimensionality reduction. We find that applying uniformity regularization significantly changes the embedding space density, which has been marked as a potential driver for generalization in Roth et al..
>
> _Roth et al., 2020, Revisiting Training Strategies and Generalization Performance in Deep Metric Learning_
>
> __5)  “One broader question that may be relevant (may not directly relate to a decision on the paper): How will uniformity on task space be affected if we leveraged a method such as Mixup [3]? How will the task space, in that case, look like?”__
>
> The effect of MixUp can be related to confidence regularization using “Vicinall Risk Minimization”, where the method works by reducing model over-confidence and undesirable oscillations when making predictions on samples differing from those encountered during training, which they achieve by training on linear combinations of images and labels.
> We are strictly different from MixUp as we encourage uniformity in the feature space of the models which improves generalization on many downstream tasks, instead of performing confidence regularization.
> MixUp is commonly used for such confidence regularization of models in semi-supervised literature.

---

> ### Author Response · Authors · 2020-11-23
> **Discussion**
>
> Kindly let us know if our response below addressed your concerns. We are happy to answer if there are additional issues/questions.

---

### Official Review · AnonReviewer1 · 2020-10-29

**Rating:** 6
**Confidence:** 3

**Review:**

In this paper, the authors claimed that uniformity in embedding space if the key for good generalization, and then propose an adversarial training based method to improve the uniformity of feature space. The claim is from previous work, thus the key contribution is the way to impose such regularization. The method itself makes sense to me.

One lacking aspect is that the authors provide no evidence on how the method works, neither quantitatively (the distance between uniform distribution and learned feature distribution) nor qualitatively (e.g. t-sne visualization on the learned feature).

Another point could improve is that I suspect the effect of uniformity is quite like imposing margin on loss function (such as AM-softmax, arcface, etc), it is better to discuss and compare with them.

I am not familiar with the dataset and SOTA performance used in evaluation. The results look reasonable to me, and could demonstrate the effectiveness of the proposed method.

Above all, I think this paper may contain some ideas that publishable, however the authors fail to dig deeper into it, and lack sufficient ablation experiments to demonstrate the method works as expected.

---

> ### Author Response · Authors · 2020-11-19
> **Response to AnonReviewer1**
>
> Thank you for your review, we have addressed each point individually.
>
> __1)  “One lacking aspect is that the authors provide no evidence on how the method works, neither quantitatively (the distance between uniform distribution and learned feature distribution) nor qualitatively (e.g. t-sne visualization on the learned feature).”... “the authors fail to dig deeper into it, and lack sufficient ablation experiments to demonstrate the method works as expected”... “this paper may contain some ideas that publishable, however the authors fail to dig deeper into it, and lack sufficient ablation experiments to demonstrate the method works as expected.”__
>
> Tab. 1 actually shows that there is evidence that as the variance of the prior increases, the model is able to perform better than the baseline. As the variance of a gaussian increases (over a limited interval, which one can realistically assume), loosely speaking, the distribution gets more “uniform”. And when we place a uniform prior, we see that the network performs the best. This further motivates our findings, showing that the uniformity in the learned posterior is key to learning a generalizable model.
> We have further provided direct comparisons with Wang & Isola on baseline meta-learning tasks in Table 2 where we see that uniformity regularization consistently performs better than the baseline on both meta-learners (which requires generalization) and across 4 datasets.
> We have also included UMAP plots of the embeddings on the training and test data in our deep metric learning where the test data is previously unseen by the model. We see that by adding uniformity regularization, improvements on the density of the embeddings can be seen, especially notable on the test data.
> This shows that uniformity regularization does indeed have an influence on the feature space.
> To further quantify this influence, we have also quantitatively evaluated the embedding space density following Roth et al. over the non-reduced features, as notable context was likely lost during the process of significant dimensionality reduction.We find that applying uniformity regularization significantly changes the embedding space density, which has been marked as a potential driver for generalization in Roth et al..
> We have also added baselines of common regularization techniques, dropout and L2 regularization, in Table 2, and continue to see consistent improvement using uniformity regularization instead.
>
> _Roth et al., 2020, Revisiting Training Strategies and Generalization Performance in Deep Metric Learning, ICML 2020_
>
>
>
>
> __2)  “Another point could improve is that I suspect the effect of uniformity is quite like imposing margin on loss function (such as AM-softmax, arcface, etc), it is better to discuss and compare with them.”__
>
> Our approach is much more fundamental than using margin-based methods in that it actively tries to map features to a Uniform Prior, and can be used alongside such methods very well and is not in direct competition (see DML experiments). We directly experiment with very related, proxy-based baselines and see improvements over them (“Softmax”, Table 3).

---

> ### Author Response · Authors · 2020-11-23
> **Discussion**
>
> Kindly let us know if our response below addressed your concerns. We are happy to answer if there are additional issues/questions.

---

### Official Review · AnonReviewer3 · 2020-10-29
**A simple and effective approach, however some comparisons are missing.**

**Rating:** 5
**Confidence:** 4

**Review:**

#### Summary

The authors propose a regularization technique that maximizes the entropy of the learned representation by regularising it with uniformity prior. The uniformity prior is imposed via an adversarial objective function.

#### Strong Points

1. The proposed regularization can be easily added to the existing frameworks to improve their generalization ability.
2. The methods improve upon the results of the baselines.
3. The paper is well-written and easy to follow.

#### Weak Points

Although the method clearly improves the results, my main concern is with the novelty of the proposed method. Secondly, the paper needs to position itself better with respect to related work which proposes more or less similar regularisations for meta-learning frameworks.

1. The idea of maximizing the entropy of the learned representation to increase its generalization capability has been explored before. In addition to the representation learning perspective in [Wang and Isola] and [2], [1] explicitly studied entropy maximization regularizations for generalization in meta-learning, although their regularization objectives are different. I would appreciate it if the authors could compare their proposed regularization with the ones in [1].

2. For uniformity regularisation, the authors propose an adversarial learning scheme. This way of inducing uniformity in the representations is also not new. For e.g. Hjelm et al [2] used the exact strategy to induce uniformity in their self-supervised representations learning method.

3. As noted by the authors, [Wang and Isola] show that the self-supervised contrastive objectives also induce uniformity in the learned representations (although in hypersphere). Such contrastive objectives have recently been used to improve transferability of meta learned representations for e.g. in [3, 4]. I would appreciate it if authors could comment on how those regularizations are different and why they should not be compared with the proposed method.


[1] Jamal et al. Task-Agnostic Meta-Learning for Few-shot Learning.

[2] Hjelm et al. Learning Deep Representations by Mutual Information Estimation and Maximization.

[3] Medina et al. Self-Supervised Prototypical Transfer Learning for Few-Shot Classification.

[4] Doersch et al. CrossTransformers: spatially-aware few-shot transfer.

---

> ### Author Response · Authors · 2020-11-19
> **Response to AnonReviewer3**
>
> Thank you for your review, we have addressed each point individually.
>
> __1)  “The idea of maximizing the entropy of the learned representation to increase its generalization capability has been explored before. In addition to the representation learning perspective in [Wang and Isola] and [2], [1] explicitly studied entropy maximization regularizations for generalization in meta-learning, although their regularization objectives are different. I would appreciate it if the authors could compare their proposed regularization with the ones in [1].”__
>
> [1] suggest to either add an entropy term to the output probabilities of the meta-learner, a setting depending on the classification nature of the specific meta-learning task at hand, or by defining “inequality measures” over each task loss to learn less task-dependent representations. The relation to entropy maximization is much less clear, the performance relies on the specific choice of inequality metric and application is still limited to a multi-task setting akin to episodic Meta-Learning. Our approach is much more generic, being able to introduce the notion of uniformity over a much larger variety of cases, independent of the specific task setup and having a clearer link to feature uniformity/max-entropy.
>
> We have adjusted the draft to reference [1] (Related Work section), and have expanded the introduction to differentiate our work more from [Wang and Isola] (see introduction). We also have added direct comparisons to Wang & Isola in table 2, where we see that adding uniformity regularization consistently outperforms Wang & Isola for both meta-learners on 4 different datasets. This is especially interesting as our regularization approach is not limited to hyperspherical representation spaces.
> We have also added baselines for common regularization techniques, dropout and L2 regularization,  in Table 2, and continue to see consistent improvement using uniformity regularization instead.
>
>
> __2)  “For uniformity regularisation, the authors propose an adversarial learning scheme. This way of inducing uniformity in the representations is also not new. For e.g. Hjelm et al [2] used the exact strategy to induce uniformity in their self-supervised representations learning method.”__
>
> Our work shares architectural similarities with Hjelm et al., who apply a similar adversarial objective to impose certain properties on the features learned through self-supervised representation learning. We note that, similar to Hjelm et al., we do not claim novelty in the adversarial optimization scheme, which was already introduced for Adversarial Autoencoders in Makhzani et al..
> However, while Hjelm et al. also matched their representations to a uniform distribution, no detailed reasons for this specific choice were given. Instead, to the best of our knowledge, our work is the first to show that it is exactly the uniformity of feature distributions introduced by our uniformity regularization that facilitate fast adaptation and transfer to novel data and tasks in neural networks, regardless of the specific application domain.
> We have included differentiation to Hjelm et al. in our Related Work section.
>
>
> _Hjelm et al.: Learning deep representations by mutual information estimation
> and maximization._
>
> _Makhzani et al.: Adversarial Autoencoders._
>
>
> __3)  “As noted by the authors, [Wang and Isola] show that the self-supervised contrastive objectives also induce uniformity in the learned representations (although in hypersphere). Such contrastive objectives have recently been used to improve transferability of meta learned representations for e.g. in [3, 4]. I would appreciate it if authors could comment on how those regularizations are different and why they should not be compared with the proposed method.”__
>
> Wang & Isola primarily state that specifically encouraging uniformity over a hypersphere is beneficial to self-supervised contrastive learning. They only theoretically show that asymptotically, optimizing contrastive losses optimizes for these uniformity quantities. Practically, Wang & Isola show that optimization of proxies to this uniformity property can further improve generalization of such contrastive approaches.
> This is similar to our motivation, in which we wish to encourage the notion of uniformity for improved generalization. However, our regularization matches a more explicit uniformity prior over *any* feature space, and as such is not limited to hyperspherical representation spaces. This opens up application to a much wider range of problem settings, which we experimentally support.
>
> Finally, we have added direct comparisons to Wang & Isola in table 2, where we see that adding uniformity regularization consistently outperforms Wang & Isola for both meta-learners on 4 different datasets.

---

> ### Author Response · Authors · 2020-11-23
> **Discussion**
>
> Kindly let us know if our response below addressed your concerns. We are happy to answer if there are additional issues/questions.

---

### Official Review · AnonReviewer4 · 2020-10-30
**Simple yet effective regularization trick with numerous applications**

**Rating:** 6
**Confidence:** 3

**Review:**

The authors argue that uniform priors for the high-level latent representations improve transferability, which is beneficial in a number of tasks involving transference. The approach is evaluated on deep metric learning, zero-shot domain adaptation and few-shot meta-learning. The authors propose a uniformity regularization term on the latent representation, implemented as an adversarial discrepancy. The results show consistent improvement in the different tasks.

Strengths
- The method is simple, yet effective.
- The paper is easy to follow and well presented.
- The approach is relevant and can be applied to multiple problems.
- The evaluation is comprehensive, showing consistent moderate gains.

Weaknesses
- The observation that uniformity helps transference was already observed by previous works, as the authors acknowledge. From that, the derivation of the regularization term is relatively straightforward.
- There is no qualitative analysis to illustrate and provide insights about why uniformity helps nor how the regularization term changes the latent representations.
- The actual methods used in different tasks are often not explained. For example, the method for the out-of-distribution task is not explained. Please provide concise yet helpful explanations.

In my opinion, the work shows an interesting contribution, but it would benefit from qualitative analysis that could provide more insights.

---

> ### Author Response · Authors · 2020-11-19
> **Response to AnonReviewer4**
>
> Thank you for your review, we have addressed each point individually.
>
>
> __1)  “The observation that uniformity helps transference was already observed by previous works, as the authors acknowledge. From that, the derivation of the regularization term is relatively straightforward”.__
>
> We do not see how the idea of aligning arbitrary feature spaces with a Uniformity Prior would follow “relatively straightforward” from the previous works. It would be appreciated if the reviewer could clarify this statement.
>
> In addition, we believe that it should not matter how difficult an idea is, much rather should it matter that, to the best of our knowledge, our paper is the first to examine the direct benefits of aligning the feature spaces with a uniform distribution prior for improved adaptation and generalization to novel tasks and data, which we further support with experiments across multiple diverse application domains.
> By directly encouraging uniformity on the features of the learned model, we set the state-of-the-art on two large scale diverse tasks in meta-learning and deep metric learning.
>
>
> __2)  “There is no qualitative analysis to illustrate and provide insights about why uniformity helps nor how the regularization term changes the latent representations.”__
>
> Thank you for the suggestion. We include some UMAP plots of the embeddings on the training and test data in our deep metric learning where the test data is previously unseen by the model. We see that by adding uniformity regularization, features are encouraged to follow a higher density distribution, which is especially visible on the test data.
> As the reduction in dimensionality can lose important high-dimensional context, we also quantitatively evaluate the embedding space density on the training data following Roth et al., for which we see a notable increase in the feature space density (30%), which in turn can be linked to improved generalization, as found by Roth et al., 2020.
>
> We have made sure to add this discussion as well as the visualizations to our draft (Section 4.6, Figure 1).
>
>
> _Roth et al., 2020, Revisiting Training Strategies and Generalization Performance in Deep Metric Learning, ICML 2020_
>
>
> __3) “The actual methods used in different tasks are often not explained. For example, the method for the out-of-distribution task is not explained. Please provide concise yet helpful explanations.”__
>
> The performance reported is the accuracy of the model, when the data is changed in a similar way as the generalization experiments in Sinha et al.. More specifically, we randomly apply the following transformations to each image on the test set to measure the generalization ability of the model: randomly translate [-4, 4] pixel, randomly rotate by [-30, 30] degrees and randomly scale by a factor of [0.75, 1.25.].
> We have updated the draft (Section 4.5) to include more detail.
>
>
> _Sinha et al.: DIBS: Diversity inducing Information Bottleneck in Model Ensembles_

---

> ### Author Response · Authors · 2020-11-23
> **Discussion**
>
> Kindly let us know if our response below addressed your concerns. We are happy to answer if there are additional issues/questions.

---

### Author Response · Authors · 2020-11-19
**Overall Comment**

We thank each of the reviewers for their time to review the paper and appreciating the simplicity of our approach, the consistent and wide-ranging applicability, our comprehensive experimentation as well as the writing of the paper!

We have addressed each reviewer individually. We have added the following to the manuscript:

* Added UMAP visualizations, as suggested by 3 reviewers, on both the training and test distributions of the data. We qualitatively see that adding regularization doesn’t hurt the learning of the training distribution, but it visually increases the distribution density of the features. We have included these visualizations in the paper (Section 4.6, Figure 1).
* We quantitatively evaluated changes in feature space density following Roth et al., where we quantitatively find an increased density as well when applying uniformity regularization. This is in line with insights made by Roth et al., which show that “an increased embedding space density is linked to stronger generalisation.” (in section 5 of Roth et al.). We have added this discussion in the paper (Section 4.6).
* Added direct comparisons with Wang & Isola on 2 meta-learners over 4 datasets. We see consistent improvements over this baseline (Table 2).
* Added two more popular regularization techniques: Dropout and L2 regularization on 2 meta-learners over 4 datasets. We see consistent improvements over both of the baselines (Table 2).
* Finally, we have added multiple clarifications in the paper and better differentiated our work from previous papers.


_Roth et al., 2020, Revisiting Training Strategies and Generalization Performance in Deep Metric Learning, ICML 2020_

---

### Decision · Program_Chairs · 2021-01-07
**Final Decision**

**Decision:**

Reject

**Comment:**

The authors argue that uniform priors for the high-level latent representations improve transferability, which is beneficial in a number of tasks involving transference. The approach is evaluated on deep metric learning, zero-shot domain adaptation and few-shot meta-learning.

Pro:
- A simple yet effective method
- Signifiant gains in experimental study

Cons:
- Close variants of this approach were proposed in previous works, and so the novelty of the current work is limited.
- There is no accompanying analysis which may shed new light on the advantages of the approach.